# The Role of Ion Channels and Intracellular Signaling Cascades in the Inhibitory Action of WIN 55,212-2 upon Hyperexcitation

**DOI:** 10.3390/brainsci14070668

**Published:** 2024-06-29

**Authors:** Sergei A. Maiorov, Denis P. Laryushkin, Kristina A. Kritskaya, Valery P. Zinchenko, Sergei G. Gaidin, Artem M. Kosenkov

**Affiliations:** Federal Research Center “Pushchino Scientific Center for Biological Research of the Russian Academy of Sciences”, Institute of Cell Biophysics of the Russian Academy of Sciences, 142290 Pushchino, Russiakosenckov406@yandex.ru (A.M.K.)

**Keywords:** cannabinoid receptors, epileptiform activity, neuronal networks, intracellular calcium, ionotropic glutamate receptors

## Abstract

Gi-coupled receptors, particularly cannabinoid receptors (CBRs), are considered perspective targets for treating brain pathologies, including epilepsy. However, the precise mechanism of the anticonvulsant effect of the CBR agonists remains unknown. We have found that WIN 55,212-2 (a CBR agonist) suppresses the synchronous oscillations of the intracellular concentration of Ca^2+^ ions (epileptiform activity) induced in the neurons of rat hippocampal neuron-glial cultures by bicuculline or NH_4_Cl. As we have demonstrated, the WIN 55,212-2 effect is mediated by CB_1_R receptors. The agonist suppresses Ca^2+^ inflow mediated by the voltage-gated calcium channels but does not alter the inflow mediated by NMDA, AMPA, and kainate receptors. We have also found that phospholipase C (PLC), protein kinase C (PKC), and G-protein-coupled inwardly rectifying K^+^ channels (GIRK channels) are involved in the molecular mechanism underlying the inhibitory action of CB_1_R activation against epileptiform activity. Thus, our results demonstrate that the antiepileptic action of CB_1_R agonists is mediated by different intracellular signaling cascades, including non-canonical PLC/PKC-associated pathways.

## 1. Introduction

Hyperexcitation of neuronal networks caused by the disturbance of the excitatory/inhibitory (E/I) balance due to excessive elevation of extracellular glutamate concentration is a distinctive feature of brain pathologies such as epilepsy, stroke, and traumatic brain injury [1,2]. A long-term increase in extracellular glutamate level results in excitotoxicity, accompanied by the dysregulation of calcium homeostasis in neurons, disruption of mitochondria function, and overproduction of reactive oxygen species, ultimately leading to cell death by triggering apoptosis or necrosis [3]. Currently, no therapeutic approach can effectively prevent excitotoxicity-induced loss of brain cells [4,5]. Moreover, an impairment in the excitation/inhibition balance causes epileptic seizures. Although many antiepileptic drugs can prevent seizures, they have several limitations and side effects and are ineffective in about 30% of cases [6,7]. For this reason, the active search for new targets that can protect brain cells from pathological hyperexcitation continues.

G-protein coupled receptors (GPCRs), especially G_i_-coupled ones, are considered the most promising targets for excitotoxicity treatment [8]. G-proteins are composed of an alpha (α) subunit and a heterodimeric complex of tightly associated β and γ subunits, often called the beta–gamma subunit (βγ). The activation of a GPCR receptor induces dissociation of α and βγ subunits, which further interact with their molecular targets. G_i_-coupled receptors are “inhibitory” since their activation is accompanied by suppression of the activity of a target cell. Generally, the α-subunit of G_i_-proteins causes a decrease in cytosolic cyclic adenosine monophosphate (cAMP) concentration via inhibition of adenylyl cyclase. In contrast, the βγ-complex directly activates G-protein-coupled inwardly rectifying K^+^ channels (GIRKs) and negatively modulates voltage-gated calcium channels (VGCCs) of the N- and P/Q-type [9]. However, these canonical effects are not universal and depend on the type of neuron and the receptor localization [10]. It should be noted that expression of GPCRs has been found not only in neurons but also in astrocytes, microglia, and other types of brain cells. For example, the activation of G_i_-coupled receptors in astrocytes may be accompanied by an increase in intra-cellular Ca^2+^ concentration ([Ca^2+^]_i_) followed by the secretion of gliotransmitters [11]. It is generally believed that [Ca^2+^]_i_ elevations occur due to phospholipase C (PLC) activation by the G_i_βγ subunit. Some studies indicate that activation of G_i_ proteins in neurons can also affect the PLC signaling pathway and the cascade involving phosphatidylinositol biphosphate (PIP_2_) and protein kinase C (PKC) [12]. The interaction of G_i_ proteins with PLC can lead to both its activation and inhibition, resulting in the change in conductivity of various channels, such as small-conductance calcium-activated potassium channels (SK channels) and glutamate receptors.

Among GPCRs, cannabinoid receptors (CBRs) are promising targets for pharmacotherapy. According to canonical classification, CBRs are divided into CB_1_ and CB_2_ subtypes. However, some new potential members of this family have been described recently, including GPCR 55 (GPR55), non-classical endocannabinoid receptors belonging to the transient receptor potential superfamily (TRP), such as the capsaicin receptor and vanilloid TRPV1 and TRPV2 channels, and nuclear peroxisome proliferator-activated receptors (PPARs), such as PPARα and PPARγ [13]. CB_1_R has the most significant representation in the brain among all GPCRs, and its expression is comparable to that of NMDA or GABA(A) receptors [14]. The main endogenous ligands of CBRs, anandamide and 2-arachidonoylglycerin, mainly act as retrograde messengers: they affect the presynaptic membrane after secretion, thus suppressing the further secretion of glutamate or gamma-aminobutyric acid (GABA). These endocannabinoids can also act on CBRs localized on the postsynaptic membrane, leading to membrane hyperpolarization [10]. In this way, the endocannabinoid system is significantly involved in the modulation of synaptic transmission in the brain, which is especially important in the case of neuropathologies.

The potential of cannabinoid receptor ligands as antiepileptic agents has been recognized since the late 1990s, when some studies unveiled the ability of CBR agonists to suppress epileptiform activity in in vitro and in vivo models [15,16]. While the antiepileptic effects of cannabinoids are widely acknowledged at present, the precise mechanisms underpinning the anticonvulsant action of CBR activation remain elusive [10,17,18]. Apparently, they go far beyond the established understanding of the work of Gi-coupled receptors. This suggests the need for further study of these extremely promising GPCRs in the context of developing new approaches for epilepsy treatment. In this study, we have tried to reveal the molecular mechanisms of the antiepileptic action of a CBR agonist, WIN 55,212-2.

## 2. Materials and Methods

### 2.1. Preparation of Hippocampal Cell Culture

Hippocampi of newborn Wistar rats (P0-2) were used to prepare neuron-glial cultures as described previously [19]. The animals were euthanized with deep-inhaled anesthesia and then decapitated. The extracted brains were placed in a sterile plastic Petri dish (60 mm diameter) filled with sterile ice-cold Versene solution. The hippocampus was carefully removed with tweezers and put into a microcentrifuge tube with cold Versene solution. After being minced with scissors, hippocampal tissue was then digested with 1% trypsin solution for 10 min at 37 °C in a thermoshaker. After that, tissue fragments were washed twice with cold Neurobasal medium and gently triturated with a pipette. The non-triturated tissue debris was removed, the cell suspension was centrifuged for 3 min at 2000 rpm, and the cell pellet was resuspended after supernatant removal in the grown medium composed of Neurobasal medium and 2% B-27. The medium was supplemented with glutamine (0.5 mM final concentration) and a penicillin–streptomycin mixture (1:250). Sterile sodium chloride solution was added to achieve the final concentration of NaCl in the grown medium of 4 g/L (concentration required for growth of postnatal neurons). The obtained suspension was seeded on round cover glasses pretreated with polyethyleneimine. The cell density was 2000 cells per 1 mm^2^ (58,000 cells per cover glass). The cultures were grown in a CO_2_ incubator (37 °C, 95% humidity, CO_2_ level 5%) for 12 days and then used in experiments.

### 2.2. Fluorescent Ca^2+^ Imaging

For the registration of changes in intracellular Ca^2+^ concentration during induced epileptiform activity, the cultures were stained with Fura-2 AM (working concentration 3 µM) for 40 min at 28 °C. The staining of cells and experiments were carried out at 28 °C in Hank’s balanced salt solution (HBSS) with the following composition: 136 mM sodium chloride, 3 mM potassium chloride, 0.8 mM magnesium sulfate, 1.25 mM potassium dihydrogen phosphate, 0.35 mM disodium hydrogen phosphate, 1.4 mM calcium chloride, 10 mM glucose, and 10 mM 4-(2-hydroxyethyl)-1-piperazineethanesulfonic acid (HEPES); pH 7.35. After that, the cells were rinsed thrice, and cover glasses with the cultures were mounted in the microscopic chambers. Fura-2 fluorescence was registered with an inverted Leica DMI 6000B microscope (Leica Microsystems, Wetzlar, Germany). We utilized an inverted Leica DMI 6000B motorized epifluorescence microscope (Leica Microsystems, Wetzlar, Germany), equipped with a Hamamatsu 9100C monochrome CCD camera (Hamamatsu Photonics K.K., Hamamatsu City, Japan). A FU-2 filter cube (Leica Microsystems, Wetzlar, Germany) with a dichroic 72100bs mirror and an HQ 540/50m emission filter was used for emission collection. Fluorescence of the probe was excited using an external Fura-2 filter wheel (Leica Microsystems, Wetzlar, Germany) with BP387/15 and BP340/20 excitation filters.

Neurons were distinguished from astrocytes by the presence of bicuculline-induced/NH_4_Cl-induced [Ca^2+^]_i_ oscillations or by KCl (35 mM) application in the case of experiments without bicuculline and NH_4_Cl (Figure 3). In turn, astrocytes were distinguished by the absence of the induced [Ca^2+^]_i_ oscillations or by short-term adenosine triphosphate (ATP, 10 μM) application in the experiments without the oscillations (Figure 3). Using the combination of vital fluorescent calcium imaging and post-vital immunostaining, we have previously shown [20] that the approach described above for the identification of neurons and astrocytes is quick and reliable.

In most experiments, the drugs were diluted to the working concentration in HBSS and were applied to the microscopic chamber using a mechanical pipette. The medium in the chamber was then gently mixed. In the case of experiments with washout (Figure 3), the drugs were added in a flow of HBSS (5 mL/min) using a perfusion system [20] to completely replace the medium. The final concentrations of the drugs in the chamber are listed in the figure legends.

### 2.3. Electrophysiology

Electrophysiological recordings were conducted in whole-cell mode at 28 °C in Hank’s Balanced Salt Solution (HBSS; see the composition in Section 2.2). The intrapipette solution employed had the following composition: 10 mM potassium chloride, 125 mM potassium gluconate, 1 mM magnesium chloride hexahydrate, 0.25 mM ethylene glycol-bis(β-aminoethyl ether)-N, N, N′, N′-tetraacetic acid (EGTA), 10 mM HEPES, 2 mM sodium adenosine triphosphate (ATP), 0.3 mM magnesium ATP, 0.3 mM sodium guanosine triphosphate (GTP), and 10 mM sodium phosphocreatine (pH 7.2; adjusted with 1 M potassium hydroxide). Data acquisition was performed using an Axopatch 200B amplifier and a low-noise Axon DigiData 1440A digitizer (Molecular Devices, San Jose, CA, USA). pCLAMP 10 software was used for data collection and analysis (Molecular Devices, San Jose, CA, USA).

### 2.4. Immunostaining

The immunostaining technique was utilized for the identification of astrocytes. For this purpose, we used a mouse monoclonal anti-GFAP (glial fibrillary acidic protein, a marker of astrocytes) antibody (see Appendix A). Fluorescence measurements of Fura-2 were conducted within one of the grid squares. Subsequent to calcium imaging, the cultures were photographed under bright-field illumination and then fixed with 4% freshly prepared paraformaldehyde for 30 min at room temperature. The cultures were then washed three times with ice-cold phosphate-buffered saline (PBS) and permeabilized using 0.1% Triton-X 100 for 10 min. Following this, we incubated the cultures with 10% goat serum (in PBS) for 30 min at room temperature to prevent non-specific binding of the secondary antibodies, after which the cultures were rinsed with 1% goat serum (in PBS). The cells were then incubated overnight at 4 °C with the primary anti-GFAP antibodies (diluted 1:200 in 1% goat serum). After this step, the cultures were washed three times with PBS and subsequently loaded with secondary goat antibodies conjugated to Alexa Fluor 647, targeting murine immunoglobulins (diluted 1:300). This step was carried out in the dark at room temperature for 90 min. Next, the cultures were washed three times with PBS and stained with 5 μg/mL of Hoechst 33342 for 5 min. The distribution of the primary antibodies was evaluated using a Leica TCS SP5 confocal microscope (Leica Microsystems, Wetzlar, Germany) within the squares where Fura-2 measurements had been performed. The fluorescence of the conjugated probe was excited using a He-Ne 633 nm laser, and emission was collected within the range of 655–700 nm. For Hoechst 33342, a UV laser with a wavelength of 405 nm was utilized, and emission was collected in the 420–500 nm range. Additionally, scans were performed in bright-field mode. Subsequently, the images of Fura-2 fluorescence and the fluorescence of the antibodies were overlaid using Adobe Photoshop Software CC 19.1.6 (Adobe Inc., San Jose, CA, USA), and changes in intracellular calcium concentration were analyzed in GFAP-positive cells.

### 2.5. Data Analysis

The analysis of images was performed using the ImageJ 1.53e software (NIH, Bethesda, MD, USA). Data analysis and statistical tests were carried out using OriginLab Pro 2016 (OriginLab, Northampton, MA, USA) and Prism GraphPad 8 (GraphPad Software, San Diego, CA, USA). Differences between groups were compared using two-way ANOVA. The variable *n* represents the number of cells analyzed in each experiment, while N denotes the number of independent repetitions. The sample size for *p* = 0.95 was calculated using OriginLab software. The required sample size for most of the datasets was ≥6. Typically, cell culture preparations from 2–3 different animals were utilized in the experiments. In most figures, representative and mean kinetics (as noted in the figure legends) are depicted. The signals are presented as a 340/387 ratio, where 340 and 387 correspond to the fluorescence intensity of the probe upon excitation with 340 nm and 387 nm light, respectively. The immunostaining images were processed using Leica LAS AF Lite software (Leica Microsystems, Wetzlar, Germany) and subsequently overlaid with the live-cell images of Fura-2 fluorescence or transmitted light images using Adobe Photoshop software.

### 2.6. Reagents

The following reagents were used in the study. (1) Sigma-Aldrich, Saint Louis, MO, USA: Paraformaldehyde (P6148), Poly(ethyleneimine) solution (Cat. no. P3143), L-Glutamic acid (Cat. no. G8415), (S)-5-Fluorowillardiine (Cat. no. F2417); (2) Life Technologies, Grand Island, NY, USA: B-27 supplement (Cat. no. 17504044), Trypsin 2.5% (Cat. no. 15090046); Goat serum New Zeland origin (Cat. no. 16210072); (3) Molecular Probes, Eugene, OR, USA: Fura-2 AM (Cat. no. F1221); (4) Tocris Bioscience, Bristol, UK: UBP 310 (Cat. no. 3621), ATPA (Cat. no. 1107), WIN 55,212-2 mesylate (Cat. no. 1038), PF 514273 (Cat. no. 3676), Go 6983 (Cat. no. 2285), ML 297 (Cat. no. 5380), Apamin (Cat. no. 1652); (5) Alomone Labs, Jerusalem, Israel: D-AP5 (Cat. no. D-145), NBQX disodium salt (N-186); (6) Cayman Chemical, Ann Arbor, MI, USA: Bicuculline (Cat. no. 11727); (7) Paneco, Moscow, Russian Federation: Neurobasal medium (H333), penicilin-streptomycin (A063); (8) AppliChem, Darmstadt, Germany: NH_4_Cl (Cat. no. 141121.1210); (9) Abcam, Cambridge, UK: U 73122 (Cat. no. ab120998), Goat Anti-Mouse IgG H&L (Alexa Fluor^®^ 647) (Cat. no. ab150115); (10) Amresco LLC, Solon, OH, USA: Triton X-100 (Cat. no. Am-O694); (11) Bialexa, Moscow, Russian Federation: mouse monoclonal antibodies to GFAP (GF-1 clone).

## 3. Results

### 3.1. Activation of CBRs Suppresses Induced Epileptiform Activity

In this work, we have investigated the effect of CB_1_R and CB_2_R receptor activation on the bicuculline- and NH_4_Cl-induced hyperexcitation of neuronal networks. Previous research has demonstrated that WIN 55,212-2 can suppress epileptiform activity in neurons [21]. As shown in Figure 1A,A′–A′′′, bicuculline application leads to the occurrence of synchronous oscillations in intracellular calcium concentration across all neurons within the network. Thus, in contrast to classical electrophysiological studies, we tested the ability of a CBR agonist to suppress the activity of not just single neurons but all neurons coupled in a united network. This network contains inhibitory GABAergic and excitatory glutamatergic neurons and astrocytes. Thus, our study shows the resulting effect of agonist application, which should consist of the activation of CBRs on various hippocampal cells. We used WIN 55,212-2, which can activate both subtypes of cannabinoid receptors. Its efficacy was examined in in vitro models of epileptiform activity triggered by the administration of the GABA(A) receptor antagonist bicuculline or by high concentrations of NH_4_Cl. The effect of the agonist was evaluated by its ability to suppress oscillations of [Ca^2+^]_i_ (calcium oscillations) that occurred synchronously in neurons during these two exposures. It has been demonstrated that NH_4_Cl induces synchronous repetitive calcium oscillations at concentrations ≥ 5 mM [22]; therefore, in the present study, we used a concentration of 8 mM as in our previous research [20]. We have found that WIN 55,212-2 effectively suppressed both bicuculline- and NH_4_Cl-induced calcium oscillations in neurons (Figure 1A,B). Preincubation with WIN 55,212-2 significantly reduces the frequency of bicuculline-induced calcium oscillations (Figure 1C). The maximum effect (almost complete absence of calcium oscillations) was observed at the agonist concentration of 500 nM (Figure 1D,E). Considering that WIN 55,212-2 can activate both CB_1_Rs and CB_2_Rs at concentrations above 50 nM, we have used PF 514273 (a CB_1_R antagonist) to determine whether CB_2_Rs are involved in the regulation of neuronal network activity (Figure 1F,G). PF 514273 completely abolished the effect of WIN 55,212-2 on bicuculline-induced epileptiform activity in two variants of the experiment: after WIN 55,212-2 application (Figure 1H) and in the case of preincubation with the antagonist followed by WIN 55,212-2 addition (Figure 1I). Thus, CB_1_Rs primarily mediate the inhibitory action of WIN 55,212-2 on neuronal network activity.

The half-maximal effect, manifested as a two-time decrease in the frequency and amplitude of [Ca^2+^]_i_ oscillations, was achieved using 50 nM of the CBR agonist (Figure 2A,A^/^). It is known that each synchronous [Ca^2+^]_i_ oscillation corresponds to the cluster of some consecutive paroxysmal depolarizing shifts (PDSs) (Figure 2B,B^/^,B^//^). Interestingly, WIN 55,212-2 (50 nM) also decreases the number of PDSs in the cluster by approximately twofold (Figure 2B,C) and hyperpolarizes the neuron. Although the decrease in the membrane potential can be short-term or long-term, the number of PDSs in the cluster changes insignificantly in this case, while the maximal membrane potential during PDS (Figure 2B^/^, PMP_max_) reduces (Figure 2C), likely due to hyperpolarization. Since the voltage-gated calcium channels predominately mediate Ca^2+^ inflow during the calcium oscillations, the fall of the oscillation amplitude after the WIN 55,212-2 application can be explained by the reduced PMP_max_.

Thus, neuronal hyperpolarization, occurring after CB_1_R activation, suppresses hyperexcitation and decreases Ca^2+^ inflow. This effect can be considered one of the factors underlying the inhibitory action of CB_1_R agonists against induced epileptiform activity.

### 3.2. The Effects of CBR Activation on the Activity of Ionotropic Glutamate Receptors and VGCCs

Next, we studied the action of WIN 55,212-2 on the activity of VGCCs and ionotropic glutamate receptors (iGluRs) (Figure 3A–E). The impact of WIN 55,212-2 was evaluated by its ability to influence the amplitudes of the calcium responses (sustained elevation of [Ca^2+^]_i_) of neurons during activation of the corresponding receptors. As shown in previous studies [23,24], domoic acid (a kainate receptor (KAR) agonist) evokes a [Ca^2+^]_i_ increase in the presence of an AMPA receptor antagonist only in neurons containing KARs with a GluK1 subunit. In this regard, a selective agonist of KARs containing this subunit was used in this work. The calcium responses of neurons to the application of FW (5-fluorowillardiine, agonist of AMPA receptors; Figure 3A), NMDA (Figure 3B), and ATPA (agonist of GluK1-containing KA receptors; Figure 3C) changed in different ways (decreased, increased, or remained unchanged) in the presence of WIN 55,212-2 (Figure 3F). We observed a significant reduction only in the amplitude of [Ca^2+^]_i_ elevation induced by KCl (Figure 3D). In the presence of NMDAR, AMPAR, and KAR antagonists, KCl-induced calcium responses occur due to the depolarization of neurons and the subsequent opening of the VGCC [21]. Therefore, the activity of the VGCC should decrease in the presence of WIN 55,212-2. The experiment with glutamate (Figure 3E) supports this hypothesis since WIN 55,212-2 results in a decrease in the amplitude of [Ca^2+^]_i_ elevation induced by glutamate, and the changes in iGluRs-mediated Ca^2+^ inflow do not underlie this effect (Figure 3A–C).

### 3.3. Contribution of PKC, PLC, GIRK, and SK Potassium Channels to the Action of CBR Activation on the Induced [Ca^2+^]_i_ Oscillations in Neurons

Next, we evaluated the contribution of GIRK (Figure 4A), SK channels (Figure 4B), PKC (Figure 4C), and PLC (Figure 4D) to the CBR-mediated suppression of neuronal network activity. Since there are currently no selective blockers of GIRK channels, their activator ML297 was used in this work. Experiments have shown that the activation of GIRK channels significantly decreases the frequency of bicuculline-induced [Ca^2+^]_i_ oscillations (Figure 4A), but complete suppression is observed only after the addition of WIN 55,212-2. These experiments suggest that GIRK channels may significantly contribute to the CBR-mediated inhibition of neuronal activity. In turn, the role of SK channels is negligible in this case (Figure 4B), since induced [Ca^2+^]_i_ oscillations were almost completely suppressed by WIN 55,212-2 in the presence of apamine (an SK channel blocker). Interestingly, PKC and PLC inhibitors (Go 6983 and U73122, respectively) abolished the effect of WIN 55,212-2 on the activity of the network: WIN 55,212-2 insignificantly affected the [Ca^2+^]_i_ fluctuation frequency in the presence of these substances. Moreover, the calcium oscillations abolished by WIN 55,212-2 occurred again when Go 6983 and U73122 were applied (Figure 4E,F). Notably, U73122 itself increased the oscillation frequency in some experiments. The results of the inhibitory analysis suggest that PKC and PLC play important roles in the signaling pathways that mediate the suppression of epileptiform activity when CBRs are activated.

## 4. Discussion

This study demonstrates that the cannabinoid receptor agonist WIN 55,212-2 effectively suppresses calcium oscillations induced by bicuculline and NH_4_Cl due to CB_1_R activation. The GABA(A) receptors form a blockade, leading to the generation of paroxysmal activity, which underlies the hyperexcitation induced by bicuculline. This paroxysmal activity is manifested in electrophysiological recordings as PDSs, which combine into clusters. Each [Ca^2+^]_i_ oscillation in a neuron corresponds to a single PDS cluster. In turn, the effect of NH_4_Cl on the cell culture is more complex since ammonium ions affect both neurons and astrocytes. As we have previously reported, [Ca^2+^]_i_ oscillations, in this case, arise due to depolarization and an increase in extracellular glutamate concentration [25,26]. WIN 55,212-2 (≥100 nM) almost immediately suppresses calcium oscillations and even restores basal [Ca^2+^]_i_ levels in the case of NH_4_Cl-induced hyperexcitation. It is known that both GABAergic and glutamatergic neurons express CB_1_Rs in the hippocampus. However, some studies report that the expression of these receptors by GABAergic neurons is more significant than expression by pyramidal neurons [27,28]. A range of works demonstrate that CB receptor agonists primarily suppress the activity of neuronal networks [29,30]. Considering that CB_1_Rs are coupled with the G_i_ protein, this effect of the CB agonists may seem paradoxical since a decrease in cAMP levels in GABAergic neurons should, conversely, lead to disinhibition of the neuronal network. However, there is evidence that despite the lower expression of CB_1_Rs in hippocampal glutamatergic neurons, the effectiveness of G-protein-mediated signaling of CB_1_Rs in these neurons is higher compared to GABAergic neurons [31]. In addition, the specific activation of CBRs at a single neuron may depend not only on the number of expressed receptors but also on the subcellular localization (soma or processes) or physiological properties of the neuron [14]. This is also indicated by the data presented in this work, according to which the addition of WIN 55,212-2 has differential effects on neuronal receptor activity. Co-expression of Cannabinoid Receptor Interacting Protein 1a (CRIP1a) and CB_1_R was found in hippocampal glutamatergic and GABAergic neurons [32,33,34]. The interaction of CRIP1a with CB_1_Rs reduces both the constitutive and agonist-dependent of the latter [35]. Moreover, the activity of the CB_1_R also depends on the interaction with other receptors, for example, with the dopamine D_2_ receptor (D_2_R), co-expression with which has been found in the hippocampus [36]. For instance, CB_1_R and D_2_R interaction can switch CB_1_R-mediated signaling from G_i/o_ to G_s_, resulting, on the contrary, in an increase in cAMP levels in neurons [37]. Furthermore, there is evidence that CB_1_Rs can interact with various alpha subunits depending on the agonist used. It has been demonstrated that in the case of WIN 55,212-2, CB_1_Rs can interact with G_αo_, G_αz_, and even G_αq_ subunits in addition to the canonical G_αi_ subunit. These changes in the interaction pattern may cause the increase in [Ca^2+^]_i_ [38]. Thus, many different factors determine the effects following CB_1_R activation. Nevertheless, in the case of the neuronal network, these receptors cause the suppression of neuronal activity, which, according to our experiments, occurs due to the decrease in VGCC activity. In addition, the almost twofold decrease in the amplitude and frequency of calcium oscillations in the presence of the GIRK channel agonist, ML 297, may indicate the involvement of these channels in the suppression of hyperexcitation. We suppose that hyperpolarization, observed in our electrophysiological experiments, is caused by the activation of GIRK. It is assumed that CB_1_Rs are localized presynaptically, where their activation attenuates the release of the neurotransmitters due to the inhibition of cAMP production and suppression of VGCC activity. However, despite the absence of anatomical evidence of CB_1_R localization at the postsynaptic terminal, their presence in the somatodendritic compartment cannot be completely excluded, as that is where they can reduce the excitability of neurons by activating potassium channels, such as hyperpolarization-activated cyclic nucleotide-gated (HCN) channels and GIRK [39,40]. An interesting result obtained in this work is that PLC and PKC inhibitors abolish and prevent the suppressive effect of WIN 55,212-2. The decision to investigate the involvement of these proteins in the attenuation of hyperexcitation induced by CBR activation was informed by our previous works [20,26], which demonstrated that the activation of α_2_-adrenergic receptors suppresses hyperexcitation induced by ammonium. This effect is achieved not only by activating neuronal receptors but also by stimulating GABA secretion by astrocytes. Furthermore, the signaling pathway leading to the release of GABA in astrocytes also involves the activation of PLC. Activation of PLC, in turn, should trigger the activation of PKC, which necessitates an increase in intracellular calcium concentration and diacylglycerol (DAG) formed during the cleavage of phosphatidylinositol 4,5-bisphosphate. Although CBRs are associated with the G_i_ protein in the classical pathway, evidence indicates that they can interact with G_q_ proteins, which may increase the level of cytosolic Ca^2+^ in astrocytes [41,42]. Moreover, PLC activation leading to an elevation in [Ca^2+^]_i_ can also occur due to interaction with the beta-gamma subunit of G_i_-protein-coupled receptors [43]. In turn, [Ca^2+^]_i_ increases in astrocytes may be accompanied by their secretion of gliotransmitters, including inhibitory ones, such as ATP and GABA [44]. Thus, analogous to the α_2_-adrenergic receptor, the activation of CBRs may suppress hyperexcitation by stimulating astrocytes. Even though in most of our experiments, we could not register an [Ca^2+^]_i_ increase in astrocytes, in some cases, astrocytes responded with a [Ca^2+^]_i_ transient to the addition of WIN 55,212-2 (see Appendix A). In other cases, the [Ca^2+^]_i_ elevation in astrocytes could be local, for example, only in their processes, which, with the resolution we used, did not allow us to detect it.

The role of [Ca^2+^]_i_ transients in the release of gliotransmitters by astrocytes is still debatable. Based on the assumption that the suppression of hyperexcitation with WIN 55,212-2 addition occurs as a result of astrocyte activation and, taking into account experiments according to which PLC and PKC inhibitors completely abolished and prevented the agonist effect, it can be assumed that PKC is a key connecting element between the activation of CBRs and the secretion of inhibitory gliotransmitters. This mechanism of hyperexcitation suppression with WIN 55,212-2 seems most likely also because, as mentioned above, CB_1_Rs on neurons are localized in large numbers presynaptically, where their activation suppresses the secretion of neurotransmitters. Within this context, the function of the PLC-PKC signaling pathway remains largely unexplored. Moreover, the mobilization of Ca^2+^ from the reticulum, in this case, should lead to stimulation of neurotransmission rather than to the opposite effect.

## 5. Conclusions

Although the inhibitory effect of CBR agonists on epileptiform activity has been recognized for an extended period, the exact mechanisms underlying it have yet to be studied. Apparently, these mechanisms are not only a classical idea of the action of G_i_-protein-coupled receptors (including, as shown in the present study, the inhibition of AP and suppression of VGCCs) but also the participation of other cell types, such as astrocytes, whose role in this effect is probably even more significant. Our results demonstrate the contribution of non-canonical PLC/PKC-mediated cascades, which may be involved in the realization of “astrocyte-dependent” anticonvulsant effects of synthetic CBR agonists or endocannabinoids. We posit that future research endeavors in this domain should concentrate on elucidating the contribution of astrocytic CBRs to the described phenomena.

## Figures and Tables

**Figure 1 brainsci-14-00668-f001:**
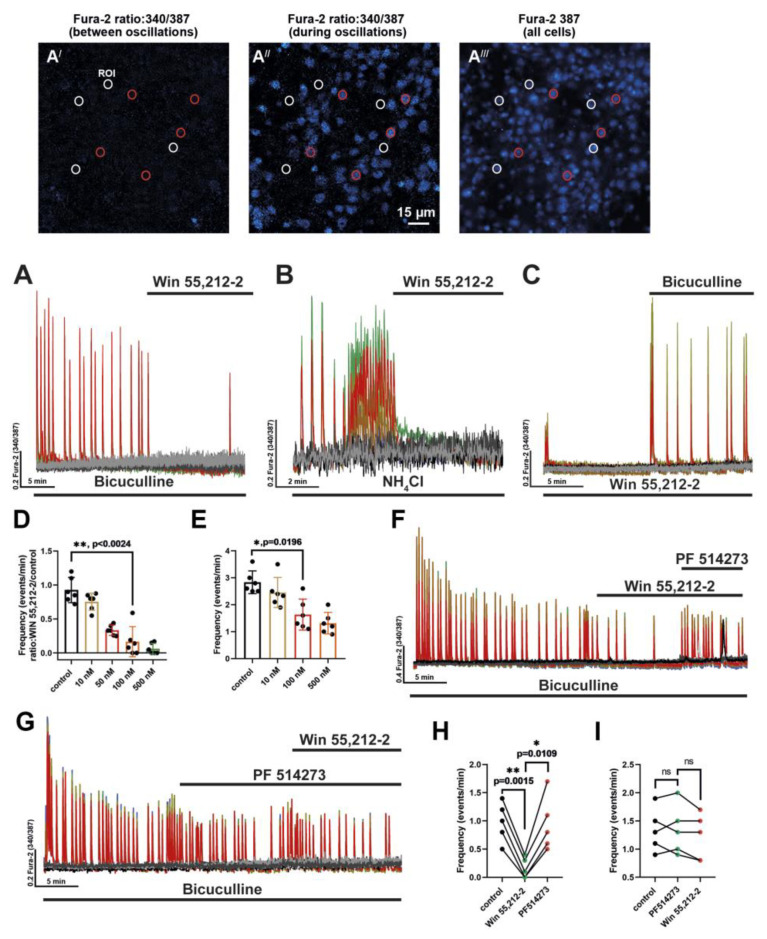
The effect of CBR activation on hyperexcitation of neuronal networks. (**A**,**B**) Application of WIN 55,212-2 (500 nM) in the presence of (**A**) bicuculline (10 μM, preincubation 10 min) and (**B**) ammonium chloride (NH_4_Cl, 8 mM, preincubation 6 min). Panels **A′** and **A″** demonstrate the images of the 340/387 ratio in the area of the neuron-glial culture between (**A′**) and during (**A″**) calcium oscillations. Panel **A′′′** shows the fluorescence of the cells upon 387 nm excitation (fluorescence of calcium-free form of Fura-2). Red circles correspond to neurons, while white circles correspond to astrocytes. (**C**) Preincubation (10 min) with WIN 55,122-2 (500 nM) before the bicuculline application (10 µM). n = 120 (100 neurons and 20 astrocytes) for experiments presented in panels **A**–**C**. (**D**,**E**) Dose-dependent changes in the frequency of [Ca^2+^]_i_ oscillations in the case of WIN 55,212-2 application after (**D**) and before (**E**) the addition of bicuculline; the differences were compared using Kruskal–Wallis analysis with the Dunn multiple comparisons test; * *p* < 0.05, ** *p* < 0.01. (**F**,**G**). The effect of PF 514273 (500 nM, CB_1_R antagonist) on the inhibitory effect of CB_1_R agonist WIN 55,122-2 (500 nM). N = 6, n = 120 (100 neurons and 20 astrocytes) neurons. Preincubation with bicuculline—15 min; preincubation with WIN 55,212-2 (**F**)—10 min; preincubation with PF 514273 (**G**)—10 min. Grey curves on all panels correspond to astrocytes, while the colored curves correspond to neurons. The representative curves demonstrating the [Ca^2+^]_i_ dynamics in individual neurons and astrocytes are shown in all panels. (**H**,**I**) The diagrams show the effect of the CB_1_R antagonist PF 514273 (500 nM) on the frequency of calcium oscillations when it is added after (**H**) or before (**I**) the addition of WIN 55,212-2 (500 nM). In diagram H: black circles—the frequency values in the presence of bicuculline only; green circles—the frequency in the presence of WIN 55,212-2; red circles—the frequency in the presence of PF 514273 (on the background of WIN 55,212-2). In diagram I: black circles—the frequency values in the presence of bicuculline only; green circles—the frequency in the presence of PF 514273; red circles—the frequency in the presence of WIN 55,212-2 (on the background of PF 514273). One-way ANOVA analysis with the Tukey test for multiple comparisons was used to evaluate the differences; * *p* < 0.05, ** *p* < 0.01. Although the application of WIN 55,212-2 did not evoke [Ca^2+^]_i_ changes in astrocytes in most experiments, we observed [Ca^2+^]_i_ transients in some cases (n = 4 for WIN 55,212-2 concentration of 500 nM). At the same time, the form of these transients varied in experiments from [Ca^2+^]_i_ oscillations to a classical metabotropic sustained biphasic [Ca^2+^]_i_ elevation (see Appendix A).

**Figure 2 brainsci-14-00668-f002:**
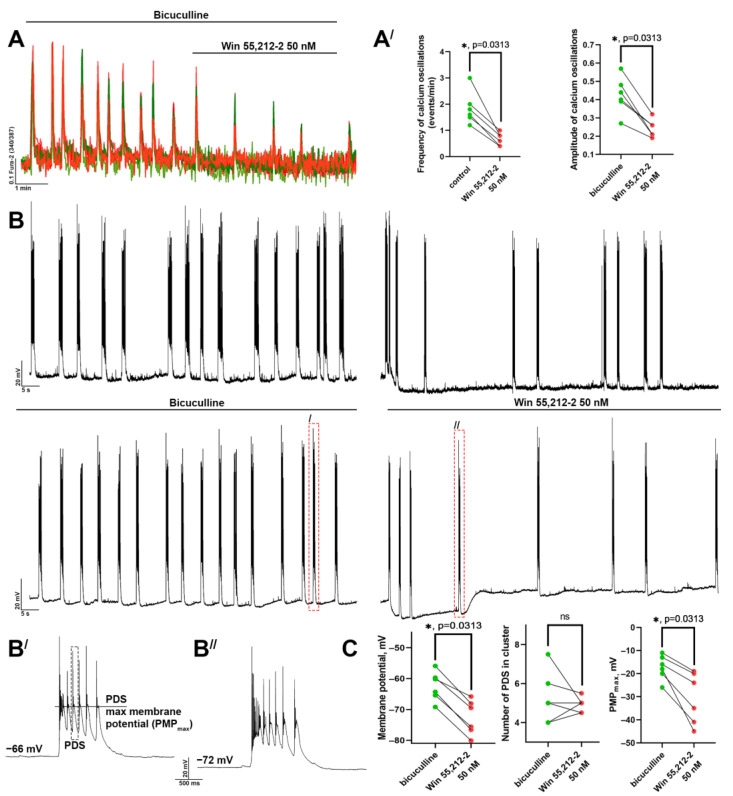
The effect of CBR activation on the electrophysiological activity of neurons. (**A**) The effect of WIN 55,212-2 (50 nM) on the (**A^/^**) frequency and amplitude of bicuculline-induced (10 μM, preincubation 10 min) [Ca^2+^]_i_ oscillations. Green and red circles show the values before and after WIN 55.212-2 application, respectively. Wilcoxon matched-pairs signed rank test was used for data comparison; * *p* < 0.05. (**B**) The epileptiform activity of neurons registered with the patch-clamp technique before and after WIN 55,212-2 application. The traces of two representative neurons with long-term (top row) and short-term (bottom row) WIN 55,212-2-induced hyperpolarization are shown. (**B^/^**,**B^//^**) Magnified images of the PDS clusters marked with red dashed rectangles in panel **B**. (**C**) The resting membrane potential, the number of PDSs in the cluster, and membrane potential during PDS (PMP_max_) before (green circles) and after (red circles) WIN 55,212-2 (50 nM) application. Differences were analyzed using the Wilcoxon matched-pairs signed rank test; * *p* < 0.05.

**Figure 3 brainsci-14-00668-f003:**
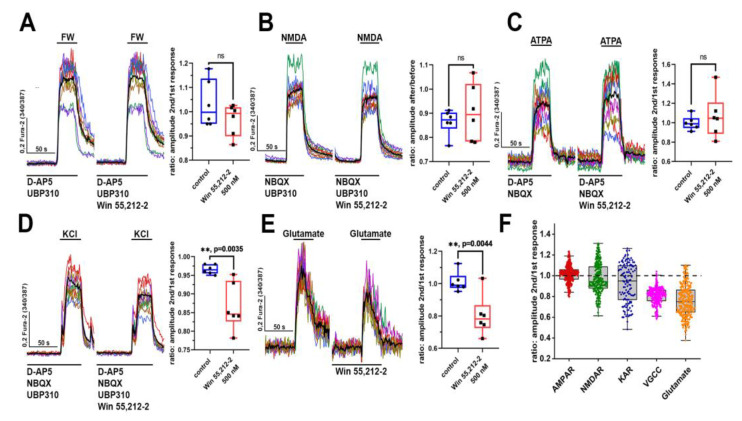
The activity of iGluRs and VGCCs in the presence of WIN 55,212-2. (**A**–**F**) The [Ca^2+^]_i_ elevations in neurons induced by the application of (**A**) NMDA (NMDA receptor agonist, 10 µM), (**B**) FW (5-fluowillardiine, AMPA receptor agonist, 500 nM), (**C**) ATPA (agonist of GluK1-containing KARs, 500 nM), (**D**) KCl (35 mM), and (**E**) glutamate (1 µM) in the presence of an appropriate combination of NMDAR, AMPAR, and KAR antagonists (D-AP5 10 µM, NBQX 2 µM, and UBP 310 10 µM, respectively). Each figure shows calcium responses of neurons before and after 1 min preincubation with WIN 55,212-2 (500 nM). The interval between the repetitive exposures was 15 min in each experiment. All the agonists (FW, NMDA, and ATPA), KCl, and glutamate were applied for 40 s. In the case of panels **A**–**D**, the cultures were incubated for 60 s with the antagonists/blockers before application of the agonist/KCl. The representative curves corresponding to the traces of individual neurons are shown in panels **A**–**E**. Bold black curves correspond to the mean curve averaged by 50 neurons in the case of the FW, NMDA, KCl, and glutamate groups. In the case of the ATPA group, the mean curve was averaged by 20 neurons in each experiment. Amplitudes of the neuronal calcium responses are shown in diagrams to the right of the graphs with [Ca^2+^]_i_ dynamics. Data are presented as the ratio of the mean response amplitude in the presence of WIN 55,212-2 to the amplitude without the CBR agonist. Control experiments were performed using a similar scheme but without WIN 55,212-2. The dots in the diagrams correspond to the mean amplitudes averaged by 50 analyzed neurons in each experiment in the case of the FW, NMDA, KCl, and glutamate groups. For the ATPA group, the dots show the mean amplitude averaged by 20 neurons in each experiment. Unpaired *t*-test; ** *p* < 0.01. (**F**) The resulting diagram showing the ratio values for experiments with WIN 55,212-2 is presented in panels **A**–**E**. Each dot corresponds to the response of an individual neuron. We analyzed 300 neurons in the case of the FW, NMDA, KCl, and glutamate groups, while in the case of the KAR group (ATPA application), the total number of analyzed neurons was 120.

**Figure 4 brainsci-14-00668-f004:**
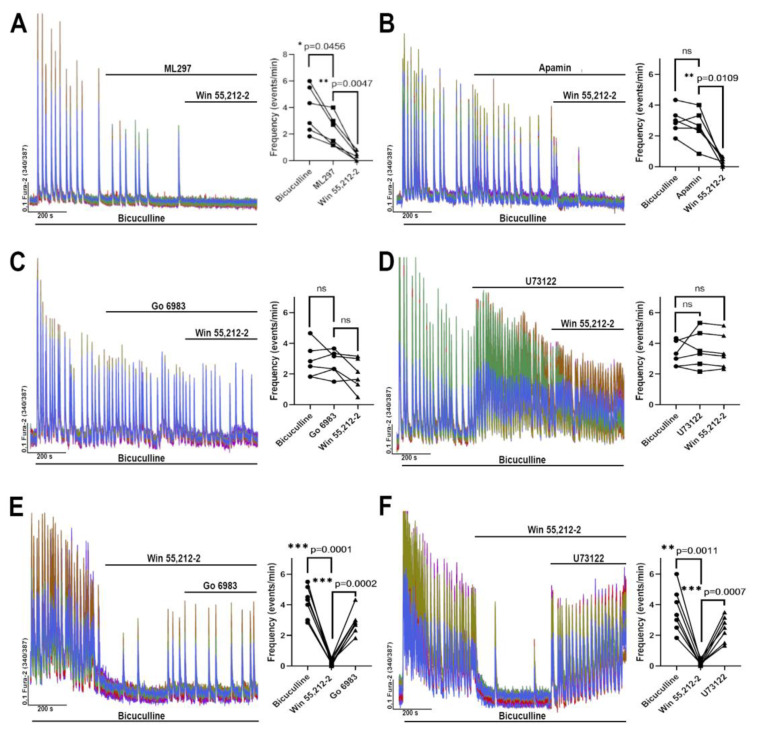
The involvement of the potassium channels, PLC, and PKC in the mechanism of WIN 55,212-2 (500 nM) inhibitory action on the epileptiform activity induced by bicuculline (10 µM, preincubation 10 min). (**A**–**D**) WIN 55,212-2 application in the presence of (**A**) GIRK activator (ML 297, 5 μM), (**B**) SK channel blocker (apamine, 100 nM), (**C**) PKC (Go 6983, 1 μM), and (**D**) PLC (U73122, 5 μM) inhibitors. Preincubation with the blockers/inhibitors—5 min. (**E**,**F**) Application of (**E**) Go 6983 (1 μM) or (**F**) U73122 (5 μM) abolishes the effect of WIN 55,212-2 on the [Ca^2+^]_i_ oscillations in neurons induced by bicuculline. Preincubation with bicuculline and WIN 55,212-2—5 min for each drug. The diagrams in panels **A**–**F** show the effect of the applied drugs on the frequency of [Ca^2+^]_i_ oscillations. n = 100. One-way ANOVA analysis with the Tukey test for multiple comparisons was used to evaluate the differences. The dots in the diagrams correspond to the mean frequency for 100 analyzed neurons in each individual experiment.

## Data Availability

The original contributions presented in the study are included in the article’s Appendix A. Further inquiries can be directed to the corresponding authors.

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
