# Peer review of "The Role of Ion Channels and Intracellular Signaling Cascades in the Inhibitory Action of WIN 55,212-2 upon Hyperexcitation"

_brainsci, 2024, doi:10.3390/brainsci14070668_

Round 1

Reviewer 1 Report

Comments and Suggestions for Authors

In the manuscript, the authors reveal that WIN 55,212-2, a CBRs agonist, suppresses the synchronous oscillations of intracellular Ca2+ ions in rat hippocampal neurons, mediated by CB1R receptors. This effect is mediated by voltage-gated calcium channels, phospholipase C, protein kinase C, and GIRK channels, indicating that the antiepileptic action of CB1R agonists is mediated by various intracellular signaling cascades.

The article is well written but minor issues should be addressed to improve it.

1.       The authors should consider including in the introduction the following paper in which he Endocannabinoid System is well described: DOI: 10.3390/cells10061282

2.       The authors should include the protocol number approved by the ethics committee

3.       The authors should include whether they performed a power analysis to calculate the actual number of Wistar rats to be used

4.       The authors should consider modifying the legend in Figure 1 as it is unclear

Author Response

Comments 1: The authors should consider including in the introduction the following paper in which he Endocannabinoid System is well described: DOI: 10.3390/cells10061282/

Response 1: We thank the Reviewer for this suggestion. This reference has been included in the introduction in the paragraph devoted to the classification of cannabinoid receptors (lines 62-68).

Comments 2: The authors should include the protocol number approved by the ethics committee

Response 2: The protocols are listed in the paragraph “Institutional Review Board Statement” at the end of the manuscript.

Comments 3: The authors should include whether they performed a power analysis to calculate the actual number of Wistar rats to be used.

Response 3: Since we usually obtain 12-14 coverslips with the cell cultures per 1 hippocampus, we have estimated the number of the required sample size similarly to our previous works (please see doi: 10.1111/jnc.15729, for example) based on the principle “1 coverslip=1 experiment”. The calculations were performed using OriginLab software. For all the presented series of experiments, P=0.95 for sample size ≥6 (≥5 for some datasets from Fig.1). 

Comments 4: The authors should consider modifying the legend in Figure 1 as it is unclear

Response 4: The legend was modified according to the suggestions of Reviewers 2 and 3. The legend is compiled to provide the reader with the maximum information necessary to understand the figure, even without referring to the main text of the article.

Reviewer 2 Report

Comments and Suggestions for Authors

1. On what basis did the authors choose to use the dose of bicuculline and NH4Cl or PF antagonist?

2. Please provide the approval number for the use of the animals.

3. The results are written in rather chaotically. For example, the authors state that WIN55,212 was used after and before the addition of bicuculline. However, there is no information regarding the time -point of its application. Therefore, please  indicate at what specific time period (minutes) before or after the bicuculline  the test compound was administered.

4. The methodology section should be rewritten and described in more detail rather than just describing the methods. The authors should clearly state how and at what doses the different compounds (depending on the purpose) were administered.

Comments on the Quality of English Language

minor changes are required

Author Response

Comments 1: On what basis did the authors choose to use the dose of bicuculline and NH4Cl or PF antagonist?

Response 1: The NH4Cl concentration was chosen based on previous works. It was shown that ammonium chloride induces repetitive calcium oscillations in neurons at concentrations≥ 5 mM. The necessary references and explanations have been added to the text. The PF 514273 concentration was chosen based on the manufacturer`s information and different studies utilizing similar objects.

Comments 2: Please provide the approval number for the use of the animals.

Response 2: The approval protocol numbers are listed in the “Institutional Review Board Statement” paragraph. We usually obtain 12-14 coverslips with the cell cultures per 1 hippocampus and rely on the principle “1 coverslip = 1 experiment”. The total number of animals used for the cell culture preparation was 15. 

Comments 3: The results are written in rather chaotically. For example, the authors state that WIN55,212 was used after and before the addition of bicuculline. However, there is no information regarding the time -point of its application. Therefore, please indicate at what specific time period (minutes) before or after the bicuculline the test compound was administered.

Response 3: We thank the Reviewer for this remark. In the case of WIN 55,212-2 application on the background of bicuculline, we performed the application 10 or 15 minutes after the bicuculline addition. Bicuculline was applied 30 seconds after the start of the experiment, and this moment is marked with the bar lines in the panels. Additional information has been added to the figure legends.

Comments 4: The methodology section should be rewritten and described in more detail rather than just describing the methods. The authors should clearly state how and at what doses the different compounds (depending on the purpose) were administered.

Response 4: We thank the Reviewer for this remark. According to the MDPI requirements, the detailed description of the previously used and routine methods has been replaced by a brief description to avoid plagiarism, including self-plagiarism. The final concentrations of the used drugs (in the microscopic chamber) are listed in the figure legends. All the drugs diluted to the working concentration in HBSS were applied to the chamber with a mechanical pipette, and the medium was gently mixed. In the case of experiments with the washout (Fig.3), the drugs were applied in the flow of HBSS (5 mL/min) to replace the medium in the chamber completely. Information about the drug application has been added to the Materials and Methods section.

Reviewer 3 Report

Comments and Suggestions for Authors

The submitted research article use a mouse cell model to investigate the role of ion channels and intracellular signaling cascades  in the activity of WIN 55,212-2 upon hyperexcitation. The main finding of this research is that different intracellular signaling, including non-canonical PLC/PKC pathways are associated to the antiepileptic and CB1 mediated action of WIN 55,212-2.

The manuscript is interesting, but needs several improvements.

Please, better describe the sectioning of hippocampus, clearly indicating the position of each cuts based on a mouse brain atlas as reference  (optimal the inclusion of a figure). How have any contaminations been avoided/eliminated?

There is no need to include the composition of standard medium, but there is the need to include information on the used antisera (i.e., the author use reference to supplementary information, but this file does not containain manufacturer, purchase code and working conditions

In result section present figure panels in alphabetical order without jumping from a panel to another (i.e., FIGURE X panel A, B, C, D, E, etc., and NOT FIGURE X panel F, G, E, A- C). Furthermore, be sure all panels within figure are described in results.

Minor

60. CB1 and CB2 are the classical cannabinoid receptors; the family of Cannabinoid receptors now includes additional GPCRs and non canonical receptors, primarly the vanilloid receptor.

Section 2.5 , as presented, is not really informative.

In methods betted describe the specific role of drugs/chemicals in the activation /supprresion of cell signaling etc.

Define abbreviations when they first appear within the main text and use consistyently all over the main text.

The relatively high iThenticate score (27%) is rescticted to material and method section only.

Comments on the Quality of English Language

fine, minor tyopos/edits need correction

Author Response

Comments 1: Please, better describe the sectioning of hippocampus, clearly indicating the position of each cuts based on a mouse brain atlas as reference  (optimal the inclusion of a figure). How have any contaminations been avoided/eliminated?

Response 1: The whole rat hippocampus (fragments from both cerebral hemispheres) was used to prepare the rat hippocampal cell cultures. Since we extracted the whole hippocampus without preliminary sectioning, the contamination by tissue from other brain regions was negligible.

Comments 2: There is no need to include the composition of standard medium, but there is the need to include information on the used antisera (i.e., the author use reference to supplementary information, but this file does not containain manufacturer, purchase code and working conditions

Response 2: Section 2.5 describes the information about the manufacturer of the primary and secondary antibodies, including the clone and catalog number. Section 2.4 describes the working dilutions of the antibodies and other details of the immunostaining procedure.

Comments 3: In result section present figure panels in alphabetical order without jumping from a panel to another (i.e., FIGURE X panel A, B, C, D, E, etc., and NOT FIGURE X panel F, G, E, A- C). Furthermore, be sure all panels within figure are described in results.

Response 3: We have reordered the references to the figures in the text and included the missing ones. We thank the Reviewer for this remark.

Minor

Comments 4: 60 CB1 and CB2 are the classical cannabinoid receptors; the family of Cannabinoid receptors now includes additional GPCRs and non canonical receptors, primarly the vanilloid receptor.

Response 4: We thank the Reviewer for this remark. We have used the well-established binary classification of the CBR family. As Reviewer 1 suggested, we have included a reference to the review article, which describes the new classification of cannabinoid receptors.

Comments 5: Section 2.5 , as presented, is not really informative.

Comments 6: In methods betted describe the specific role of drugs/chemicals in the activation /supprresion of cell signaling etc.

Response 5 and 6: The biological effects of each used agonist, antagonist, and inhibitor are mentioned in the main text and indicated in the figure legend to provide the required information for readers without needing to refer to the materials and methods section. To prevent overload of the main text with the manufacturer’s names, full names of the reagents, and catalog numbers, we have included this information in a separate section.

Comments 7: Define abbreviations when they first appear within the main text and use consistyently all over the main text.

Response 7: The authors thank the Reviewer for this remark. Additional abbreviations have been added to the text.

Comments 8: The relatively high iThenticate score (27%) is rescticted to material and method section only.

Response 8: According to the reply from Section Managing Editor Ms. Simone Liu, the above-mentioned similarity score is acceptable for the materials and methods section.

Round 2

Reviewer 2 Report

Comments and Suggestions for Authors

The authors provided sufficient explanations and corrections to the text, thus improving the paper. In my opinion, it can be published in the present form